# 3-Aminopyridine Salicylidene: A Sensitive and Selective Chemosensor for the Detection of Cu(II), Al(III), and Fe(III) with Application to Real Samples

**DOI:** 10.3390/ijms232113113

**Published:** 2022-10-28

**Authors:** Yousef M. Hijji, Rajeesha Rajan, Amjad M. Shraim

**Affiliations:** Department of Chemistry and Earth Sciences, College of Arts and Sciences, Qatar University, Doha PO. Box 2713, Qatar

**Keywords:** Schiff base, sensors, salicylidimines, fluorescence, aluminum, iron, copper, antiperspirant, secret writing

## Abstract

Interest in developing selective and sensitive metal sensors for environmental, biological, and industrial applications is mounting. The goal of this work was to develop a sensitive and selective sensor for certain metal ions in solution. The goal was achieved via (i) preparing the sensor ((E)-2-((pyridine-3-ylimino)methyl)phenol) (3APS) using microwave radiation in a short time and high yield and (ii) performing spectrophotometric titrations for 3APS with several metal ions. 3APS, a Schiff base, was prepared in 5 min and in a high yield (95%) using microwave-assisted synthesis. The compound was characterized by FTIR, XRD, NMR, and elemental analysis. Spectrophotometric titration of 3APS was performed with Al(III), Ba(II), Cd(II), Co(II), Cu(II), Fe(III), Mn(II), Ni(II), and Zn(II). 3APS showed good abilities to detect Al(III) and Fe(III) ions fluorescently and Cu(II) ion colorimetrically. The L/M stoichiometric ratio was 2:1 for Cu(II) and 1:1 for Al(III) and Fe(III). Low detection limits (μg/L) of 324, 20, and 45 were achieved for Cu(II), Al(III), and Fe(III), respectively. The detection of aluminum was also demonstrated in antiperspirant deodorants, test strips, and applications in secret writing. 3APS showed high fluorescent selectivity for Al(III) and Fe(III) and colorimetric selectivity towards Cu(II) with detection limits lower than corresponding safe drinking water guidelines.

## 1. Introduction

Due to the essentiality and toxicity of metal ions to living organisms, there is a great interest in developing sensitive and selective chemical sensors for the detection and quantification of metal ions in environmental and biological matrices. Among the metal ions that are of interest in this contest are aluminum (Al(III), copper (Cu(II), and iron (Fe); the three metal ions that are selectively sensed by 3APS in the current work.

Aluminum is the third most abundant metal on the earth’s crust and is widely used in domestic and industrial structural components, such as electrical and electronic devices, household utensils, and packaging and building materials. It is also used in common medications, such as antiulcer and antacids, and in antiperspirants. As a result of its wide usage, aluminum has already caused environmental contamination mainly from industrial waste discharge, which results in a high degree of soil and water pollution due to leaching via acid rain [1,2,3]. Exposure to aluminum causes neurotoxicity and many other health complications, including Alzheimer’s disease [4,5]; respiratory, bladder, and breast cancers [6,7]; toxicity and iron deficiency in infants and genotoxicity [8,9]; cognitive impairment [10]; and kidney and bone diseases [11,12,13]. Consequently, safe limits have been introduced for aluminum in drinking water. For example, the maximum acceptable concentration for total aluminum in drinking water in Canada is 2.9 mg/L [14], and a secondary maximum contamination limit of 0.05 to 0.2 mg/L has been established by the USEPA [15].

Copper is one of the major micronutrients in the human body that is vital for the formation of red blood cells. It is also important for bone growth, nerves, the immune system, and iron absorption. While low levels of Cu(II) are known to cause metal deficiency, high Cu(II) concentrations may induce toxicity. For instance, ingesting excessive concentrations of Cu(II) causes gastrointestinal disturbance, and prolonged exposure can cause liver and kidney damage [16,17,18,19]. Copper is also known to cause toxicity in plants [18,20].

Iron is an essential element for the human body growth and development, particularly at early stages of growth, as it makes an integral part of hemoglobin, myoglobin, and several hormones. Exposure to excessive concentrations of iron in humans is highly toxic and leads to many health problems, such as organ damage [21], oxidative stress and neurodegeneration [22], and several types of cancers [23,24]. On the other hand, iron deficiency is a leading cause of diseases worldwide, particularly in low-income countries. Anemia, for example, is a major manifestation of iron deficiency, especially in children, which is associated with several types of behavioral and cognitive abnormalities. It is also associated with heart failure and several other diseases [25,26,27,28,29].

Therefore, it is important to develop simple and cost-effective detection methods for metal ions in environmental and biological samples. However, it has been a challenge for chemists to design and synthesize chemosensors that can efficiently sense metal ions of biological and environmental relevance, such aluminum, zinc, iron, lead, mercury, and copper. Current literature is witnessing increased interest in developing selective metal probes [30,31,32]. Many fluorescent molecular probes have been developed for the detection of metal ions, such as azo arm functionalized sensors for Al(III) [33], FRET amplifier chemosensors for Al(III) [34], polyfluorene sensor for the detection of Fe(III) [35], aminopropyl-functionalized mesoporous material for Zn(II) [36], and merocyanine, a polymethine dye, for Cu(II) [37]. Interestingly, dual metal ion colorimetric and fluorescent chemosensors have also been developed for Cu(II) [38] and Al(III) [39,40,41]. The julolidine and imidazole imine colorimetric sensor for Fe(III)/Fe(II) and the fluorimetric sensor for Al(III) [42] and a series of bishydrazide compounds as Al(III) and Zn(II) sensor [43] have also been reported.

Even though extensive work has been reported in the literature for developing chemical sensors for metal ions, there is always a demand on synthesizing more efficient and accessible sensors with wider application and specific properties. In here, we are reporting a simple, easy, and fast method to synthesize a chemical probe capable of detecting aluminum and iron ions using fluorescence and copper ions with UV–VIS absorption spectroscopy. The detection of aluminum was demonstrated in real cosmetic samples.

## 2. Results and Discussion

### 2.1. Synthesis of 3APS

The compound was synthesized efficiently under microwave irradiation in just 5 min (Figure 1) with an excellent yield (95%). The chemical structure of the compound was confirmed using FTIR, ^1^H-NMR, ^13^C-NMR, and XRD, as shown in Figure 1. In fact, the preparation of 3APS has already been reported in the literature based on conventional methods similar to that reported by Smith and Warren [44], which requires reflux and long times to prepare. For instance, the compound was synthesized via condensation of 3-aminopyridine with salicylaldehyde and heating for 15 min over a steam plate and leaving the reaction overnight for the product to appear [45] or refluxing for 1 h [46]. The preparation of similar compounds (i.e., 2-(3-pyridylmethyliminomethyl)phenol) required refluxing for 2 h [47]. Therefore, the method of synthesis that is developed in the current work (Section 3.2) is superior to the methods already reported in the literature due to the short synthesis time (5 min compared with 15–60 min heating or refluxing) without the need for waiting overnight for the product to appear as reported in the literature.

### 2.2. Absorption Behavior of the 3APS Sensor in Different Solvents

To study the solvent effect on the absorbance of the sensor, solutions of 1.0 × 10^−^^3^ M of 3APS in ethanol, acetonitrile (ACN), methanol, and water were prepared, and their absorbance spectra were measured. The compound showed similar absorbance profiles in all investigated aprotic polar solvents. However, due to the low solubility of 3APS in water, its aqueous solution showed very low absorbance over the investigated wavelength range. For environmental applications, it is crucial to develop sensors that work in aqueous media; therefore, the use of H_2_O–ACN (9:1) as a mixed solvent was adopted.

### 2.3. UV–VIS Studies on 3APS as a Metal Sensor

A known volume of the 3APS stock solution (2 × 10^−^^4^ M) in ACN was mixed with a known volume of each of the following metals: Al(III), Ba(II), Cd(II), Co(II), Cu(II), Fe(III), Fe(II), Mn(II), Ni(II), and Zn(II) (1 × 10^−^^3^ M in water). The final concentrations of 3APS and each of the metal ions were kept at 2 × 10^−5^ M and 1 × 10^−4^ M, respectively. The final solutions of the 3APS–metal mixtures were diluted with water or ACN to achieve the desired final H_2_O–ACN ratio (i.e., 9:1 or 1:9). The absorbance was taken at 1 min, and the spectra are shown in Figure 2. No absorbance was observed after 500 nm. The selectivity of 3APS to metal ions was found to be solvent ratio dependent. Interestingly, an aqueous medium of 90% water showed good selectivity towards Cu(II) at 385 nm (Figure 2a). Moreover, Al(III) and Fe(III) displayed an absorption band at 250 nm. On the other hand, 3APS in 90% organic phase showed no selectivity towards any of the tested metals (Figure 2b).

The interaction of metal ion solutions with 3APS in H_2_O–ACN (9:1) was also visually examined. The results are displayed in Figure 3a (metal solutions only) and Figure 3b (metals with 3APS). Only Fe(III) gave a detectable color change after mixing with 3APS. Fe(III) is known to give a violet color with the phenolic hydroxyl group, while Al(III) decolorizes the pale-yellow color of 3APS.

### 2.4. UV–VIS Titration of Cu(II) against 3APS

To investigate the interaction of Cu(II) with 3APS, the sensor solution (2 × 10^−^^5^ M) was titrated with varying concentrations of Cu(II) solution (5–1000 µM) in H_2_O–ACN (9:1), and the spectra are shown in Figure 4a. A new peak at 380 nm is observed with increasing Cu(II) concentration, and the original 336 nm peak decreased subsequently, a sign of forming the 3APS–Cu(II) complex. The titration of Cu(II) was studied using the Benesi–Hildebrand method, and the binding constant was calculated to be 6.5 × 10^3^ M^−1^, as shown in Figure 4b. The limit of detection (3.3σ/slope) of Cu(II) in H_2_O–ACN (9:1) [48] was found to be 0.12 μM (0.324 mg/L), which was lower than the USEPA’s allowed limit of copper in drinking water (1.3 mg/L) [49].

### 2.5. Job’s Plot for the 3APS-Cu(II) System

The stoichiometric ratio of Cu(II) to 3APS was calculated from Job’s plot, as shown in Figure 5a. Solutions of equal concentrations (1 × 10^−^^4^ M) of Cu(II) in water and 3APS in ACN were mixed at varying ratios (9:1, 8:2, 7:3, 6:4, 5:5, 4:6, 3:7, 2:8, 1:9, and 0:10) to produce a final concentration of 1 × 10^−^^5^ M, and the absorbance of each solution was measured at 385 nm. The apex indicates a stoichiometric ratio of 2:1 between 3APS and Cu(II). The possible structure for the complex is presented in Figure 5b, where Cu(II) binds the two 3APS molecules at the phenolic–OH groups and the imine Ns, as reported by Wang and Zheng [50]. We were unable to obtain crystals of the complex, possibly due to the solubility of the complex in water, causing it not to crystalize. This proposed structure for the complex is mainly based on Job’s plot and published data [50].

### 2.6. Interference of Other Metal Ions with 3APS–Cu(II)

To investigate the interference caused by other metal ions with the 3APS–Cu(II) complex, solutions of each metal ion were added individually to the 3APS–Cu(II) mixture at a molar ratio of 1:2, and the absorbances were measured (see Figure 6). All metal ions did not show considerable interference with the 3APS–Cu(II) complex except Al(III) and Fe(III), where they reduced the absorbance by less than half. This reduction might be due to the stronger binding of 3APS with Al(III) or Fe(III).

### 2.7. Fluorescence of the 3APS–Metals System

A solution of 3APS (2 × 10^−^^4^ M) in ACN was mixed with various metal solutions (1 × 10^−^^3^ M) in deionized water, one at a time to produce solutions containing 2 × 10^−^^6^ M 3APS and 1 × 10^−^^4^ M of each metal in H_2_O–ACN (9:1). The emission for each solution was measured at an excitation wavelength of 350 nm. It is obvious that the intensity of 3APS at 403 nm enhanced greatly (sevenfold) after the addition of Al(III), followed by Fe(III) (sixfold), Fe(II) (fourfold), and Cu(II) and Co(II) (about twofold each). The rest of the metals did not provide any appreciable enhancement, as shown in Figure 7a. The fluorescence of the 3APS–metal interactions was also visually taken at 365 nm, as shown in Figure 7b,c. Apparent fluorescence was observed upon the addition of Al(III), Cu(II), Fe(III), and Fe(II). It is worth mentioning that 3APS was employed earlier [51] as a sensor for the detection of Ba(II) in the presence of several metal ions similar to those investigated in the current work. However, the 3APS–metal mixtures investigated in Zhao et al. [51] were buffered at pH 7.0 using HEPES (4-(2-hydroxyethyl)-1-piperazineethanesulfonic acid). Discrepancies between our results and the reported one could be explained by the effect of buffer on the binding ability of 3APS to different metal ions.

### 2.8. Fluorimetric Titration of 3APS with Al(III)

A solution of 3APS (2 × 10^−^^6^ M each) was titrated with varying concentrations of an Al(III) solution (1.7 × 10^−^^7^ M–1× 10^−^^4^ M) at an excitation wavelength of 350 nm. The fluorescence intensity of 3APS at 405 nm enhanced gradually upon adding Al(III) and reached a saturation level at a concentration of 9 × 10^−5^ M, as shown in Figure 8a. The 3APS binding constant with Al(III) was determined using the Benesi–Hildebrand method and found to be 3.2 × 10^4^ M^−^^1^, as shown in Figure 8b. The limit of detection (3.3σ/slope) [52,53] for Al(III) was found to be 0.75 μM (0.020 mg/L), which is much lower than the maximum acceptable concentration for total aluminum in drinking water in Canada (2.9 mg/L) [14] and the USEPA secondary maximum contamination limit (0.05 to 0.2 mg/L) [15].

### 2.9. Job’s Plot for 3APS–Al(III) Interaction

To further assess the coordination stoichiometry of 3APS with Al(III), solutions of 3APS in ACN and Al(III) in water of equal concentration (2 × 10^−^^4^ M) were mixed in varying ratios of 9:1–0:10 to obtain a final concentration of 5 × 10^−^^6^ M. Job’s plot indicates a ratio of 1:1 for the 3APS: Al(III) complex as illustrated in Figure 9a and as reported by others for the complexation of Al(III) with other tridentate ligands [42]. The proposed structure of the complex could be formed from two 3APS molecules binding two Al(III) molecules, as demonstrated in Figure 9b. This binding is believed to occur through the phenolic–OH group and the imine N in the first 3APS molecule and the pyridine N of the other 3APS molecule with the first Al(III) atom. The same would occur for the other Al(III) atom to satisfy the oxidation number of Al. We attempted to synthesize the complex but were unable to get crystals for X-ray.

### 2.10. Interference of Other Cations on the Interaction of 3APS with Al(III)

To evaluate the interference of other metal ions with the 3APS–Al(III) interaction, the fluorescence response of 3APS to various metal ions in the presence of Al(III) in H_2_O–ACN (9:1) was investigated, and the results are shown in Figure 10a. The concentration of each of the other metal ions added to the 3APS–Al(III) solution was 10-fold higher. Only minimal interferences in the 3APS–Al(III) emission intensity were observed from all metals (±10%), except Ni(II) (reduction in intensity) and Fe(III) (enhancement in intensity) with about 20% interference, while Cu(II) showed about 35% interference as intensity reduction. This effect may be due to the similar coordination behaviors of the later three metal ions (i.e., Ni(II), Fe(III), and Cu(II)) [54]. The interference of various metal ions with 3APS–Al(III) interaction was also visually investigated under UV lamp at 365 nm, as illustrated in Figure 10b, which exhibits a clear visual effect for Fe(III) on the 3APS–Al(III) complex.

### 2.11. Fluorimetric Titration of 3APS with Fe(III)

Solutions of 3APS (2 × 10^−^^6^ M each) were titrated with Fe(III) solutions of varying concentrations (3.3 × 10^−^^7^ M–6 × 10^−^^5^ M) at λ_ex_ 405 nm. Upon adding Fe(III), the fluorescence intensity of 3APS increased from its initial value to reach a saturation level at 5.0 × 10^−^^5^ M and started to diminish afterwards, as shown in Figure 11a. The Benesi–Hildebrand plot was developed, and the binding constant was calculated to be 1.25 × 10^5^ M^−^^1^, as shown in Figure 11b. The limit of detection (3.3σ/slope) [52,53] for Fe(III) turned out to be 1.19 µM (0.0066 mg/L), which is lower than the USEPA secondary maximum contamination limit (0.300 mg/L) [15]. To further clarify the coordination stoichiometry of 3APS with Fe(III), Job’s plot using fluorescence data was developed, and a ratio of about 1:1 was observed, as shown in Figure 11c. Similar findings were reported by another researcher [42]. A possible complex structure could be with the binding sites as the phenolic–OH group, the imine N and the pyridine N, as suggested earlier for Al(III) to satisfy the oxidation state of Fe(III), as shown in Figure 11d. Attempts to synthesize the complex were unsuccessful, and no crystals were obtained to confirm the structure by X-ray.

### 2.12. Interference of Other Metal Ions on Interaction of the 3APS–Fe(III) Complex

The fluorescence response of 3APS solutions (2 × 10^−^^6^ M each) to Fe(III) (1 × 10^−^^6^ M) before and after the addition of various metal ions (1.0 × 10^−^^5^ M each) in H_2_O–ACN (9:1) was investigated, and the results are shown in Figure 12. The fluorescence was recorded at λ_ex_ 350 nm and λ_em_ 405 nm. Apart from Al(III), the 3APS sensor showed high selectivity for Fe(III) in the presence of other metal ions, where the interference was ~5%.

### 2.13. Application of 3APS to Detect Aluminum in Cosmetic Products

To demonstrate the capability of 3APS as a chemosensor for Al(III), the sensor was employed in several real samples, including cosmetic products, secret writing, and alum crystal (as prepared in Section 2.14).

We chose four bars of antiperspirant deodorants (C, D, E, F) and two antiperspirant dry sprays (G and H), which are commercially available in Qatar, of which D was labelled as containing aluminum zirconium tetrachlorohydrex GLY 15.2%, F contained 19.2% Al, G and H contained aluminum chlorohydrate, and C and E were aluminum free. To test the applicability of employing 3APS as a sensor for Al(III) in these commercial products, 16 strips of Schleicher and Schuell ashless filter papers were grouped into two sets. For the first set (Set I, Figure 13a), each sample was applied to a different strip. A blank strip containing only 3APS and a control strip coated with Al(III) solution were also used. The same application was performed to the other set but with 3APS applied to each strip (Set II, Figure 13b). The two sets were visualized under UV lamp at 365 nm. It is obvious that the samples containing aluminum (D, F, G, and H) showed blue fluorescence, while the Al-free samples (C and E) showed no fluorescence. The control strip (B), which is 3APS–Al(III) coated, also showed blue fluorescence.

3APS was also utilized in secret writing by applying its solution on a filter paper and allowed to dry. Then aluminum solution was used to write on the 3APS-coated filter paper and visualized under normal light (Figure 14a) and under UV light at 365 nm (Figure 14b). While the application of the Al(III) solution displayed a blue fluorescence under UV light (Figure 14b), no fluorescence was observed under normal light (Figure 14a).

### 2.14. Synthesis of 3APS–Alum Crystals

Two grams of aluminum foil was dissolved in 1.5 M NaOH solution by heating. Hydrogen gas was released, and the obtained solution was filtered while hot. To the filtrate, 300 mg of 3APS was added, and the solution turned yellow as the 3APS encountered deprotonation in basic conditions. On neutralization with 5 mL of 9 M H_2_SO_4_, the yellow color diminished, and a fluorescent creamy precipitate formed, as shown in Figure 15a, while alum did not show any fluorescence (Figure 15b). Large crystals were formed with the one preincorporated with 3APS showing fluorescence under UV lamp at 365 nm, while the one with no 3APS added did not fluoresce (Figure 15c).

## 3. Materials and Methods

### 3.1. Reagents and Instruments

O-salicylaldehyde; 3-amino pyridine; ethanol; nitrate salts of Al(III), Ba(II), Cd(II), Co(II), Cu(II), Fe(III), Mn(II), Ni(II), and Zn(II), and sulfate salt of Fe(II) were all purchased from Sigma-Aldrich (Steinheim, Germany) and dissolved in ultrapure water (Direct-Q 5 UV water purification system, Millipore, Burlington, MA, USA). The microwave reactions were carried out in a Biotage Initiator system (Biotage, Sweden), the temperature and time were preset as required, and the pressure was monitored and indicated. The identity of the products was determined by FTIR and NMR. FTIR analysis was performed on a Spectrum BX FTIR spectrometer (PerkinElmer, Waltham, MA, USA). The ^1^H- and ^13^C-NMR spectra were recorded on a JNM-ECZR series 600 MHz spectrometer (JEOL, Tokyo, Japan) operating at 600 and 100 MHz, respectively, using DMSO-d6 as a solvent. The NMR is equipped with a Delta™ NMR data processing software. UV–VIS spectra were carried out using an Agilent 8453 spectrophotometer (Agilent, Santa Clara, CA, USA) in a 1.0 cm quartz cuvette. Fluorescence studies were performed on a PerkinElmer LS 45 Fluorescence spectrometer (PerkinElmer, Waltham, MA, USA) using a 1.0 cm quartz cuvette with a scan speed of 700 nm min^−1^, and an excitation and emission slit of 10 nm each.

#### Stock Solutions of the Sensor (3APS)

(E)-2-((pyridine-3-ylimino) methyl) phenol, synthesized as indicated in the experimental section (Section 3.2), was prepared in ACN at a concentration of 2 × 10^−^^4^ M. Stock solutions of metal ions as cations were made in ultrapure water (1.0 × 10^−^^3^ M). UV–VIS measurements were carried out using the 3APS sensor (2 × 10^−^^5^ M) alone and the sensor with metal ions (1.0 × 10^−^^3^ M in H_2_O: ACN, 9:1), one at a time. Fluorescence measurements were performed in a similar way as absorbance but with the 3APS sensor at 2.2 × 10^−6^ M and varying concentrations of metal ions at an excitation wavelength of 350 nm.

### 3.2. Synthesis

(E)-2-((pyridine-3-ylimino) methyl) phenol (3APS) was prepared as follows: An amount of 1.97 mmol of 3-aminopyridine was dissolved in 5.0 mL of absolute ethanol inside a microwave glass vessel, followed by the addition of 1.2 eq of o-salicylaldehyde. The mixture was microwaved for 5 min at 150 °C. After removal of the solvent in vacuo, the residue was cooled to room temperature, obtaining a yellow solid. Pure 3APS was obtained (0.37 g) after recrystallizing twice from absolute ethanol (95% yield). ^1^H-NMR (DMSO, TMS): δ 12.59 (s, 1H), 9.02 (s, 1H), 8.63 (d, J = 2.4 Hz, 1H), 8.51 (d, J = 4.65 Hz, 1H), 7.88 (t, J = 9.29 Hz, 1H), 7.69 (d, J = 7.83 Hz, 1H), 7.52–7.43 (2H), 7.01 (t, J = 7.58 Hz, 2H). ^13^C-NMR (DMSO, TMS) 165.62 (1C-O, benzene), 160.61 (1C-imine), 148.20 (1C, -N=C, 3-pyridine), 145.00 (1C, imine-N, 3-pyridine), 143.96 (1C, 3-pyridine), 34.23 (1C, benzene), 132.96 (1C, benzene), 128.41 (1C, pyridine), 124.64 (1C, pyridine), 119.86 (1C, benzene), 119.79 (1C, benzene), 117.17 (1C, benzene). IR: 1613 cm^−1^ imine group (-C=N), 1568 cm^−1^ (C=C), 1476.84 cm^−1^ (-C-N-), 1283 cm^−1^ group (-C-O-), and OH have very low broad absorption at 2000–3500 cm^−1^. Anal. Calcd for C_12_H_10_N_2_O (198.08): C, 72.71%; H, 5.08%; N, 14.13%. Found: C, 72.72%; H, 4.08%; N, 14.39%. The structures and synthetic route are shown in Figure 1.

## 4. Conclusions

The salicylidene 3-aminopyridine Schiff base was synthesized quickly (five min) and efficiently (95% yield) using MW, and its selectivity towards various metal cations was evaluated. It showed high selectivity towards Cu(II) using UV–VIS in H_2_O–ACN (9:1) and high selectivity for Al(III) and Fe(III) ions using fluorescence. The stoichiometric ratio between 3APS and Cu(II) was found to be 2:1, while it was 1:1 for Al(III) and Fe(III). The lowest detection limits for the tested ions were in the micromolar levels, which were lower than safe drinking water guidelines. The probe was applied for the fluorimetric detection of aluminum in a solution and in commercial antiperspirant products and alum crystals. Moreover, the sensor was applied in secret writing.

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
