# Peer review of "3-Aminopyridine Salicylidene: A Sensitive and Selective Chemosensor for the Detection of Cu(II), Al(III), and Fe(III) with Application to Real Samples"

_ijms, 2022, doi:10.3390/ijms232113113_

Round 1

Reviewer 1 Report

Manuscript ID: Manuscript ID: ijms-1991788 Type of manuscript: Article Title: 3-Aminopyridine salicylidene: a sensitive and selective chemosensor for the detection of Cu(II), Al(III), and Fe(III) with application to real samples  

Dear Editor!

The authors submitted a manuscript about a salicylidene 3-aminopyridine Schiff base which was synthesized quickly and efficiently (95% yield) using MW and its selectivity towards various metal cations were evaluated. It showed high selectivity towards Cu(II) using UV-Vis in H2O-ACN (9:1) as well as high selectivity for Al(III) and Fe(III) ions using fluorescence. The stoichiometric ratio between 3APS and Cu(II) found to be 2:1 while it was 1:1 for Al(III) and Fe(III). The  lowest detection limits for the tested ions were in the micro-molar levels which were lower than safe drinking water guidelines. The probe was applied for the fluorimetric detection of aluminum in solution as well as in commercial antiperspirant products and alum crystals.

Having carefully read the text, I can give a positive review of this manuscript. I believe that in this form the article can be published in such a competent journal as Materials (Q1).

My recommendations are as follows.

1. The publication abstract needs to be rewritten. This section of the article should contain a summary of the results of the study and its objectives. In its current form, the text of the abstract contains only vague phrases. It should be more informative.

2. Drawings (Fig. 5 and 9)  need to be redone.

Author Response

Many thanks for the reviewers for the time and efforts they took to revise the manuscript.

Reviewer #1

Comment #1. Having carefully read the text, I can give a positive review of this manuscript. I believe that in this form the article can be published in such a competent journal as Materials (Q1).

Authors reply #1: many thanks for the reviewer for his careful review and positive comments.

Comment #2. The publication abstract needs to be rewritten. This section of the article should contain a summary of the results of the study and its objectives. In its current form, the text of the

abstract contains only vague phrases. It should be more informative.

Authors reply #2: due to the 200 word limit requested by the editor, we couldn’t include

more results in the abstract. However, we made adequate changes to address the reviewer’s comment. The revised Abstract is:

Background: interest in developing selective and sensitive metal sensors for environmental, biological, and industrial applications is mounting. The goal of this work was to develop a sensitive and selective sensor for certain metal ions in solution. The goal was achieved via (i) preparing the sensor ((E)-2-((pyridine-3-ylimino)methyl)phenol) (3APS) using microwave radiation in a short time and high yield, and (ii) performing spectrophotometric titrations for 3APS with several metal ions. Methods: 3APS, a Schiff base, was prepared in five minutes and in a high yield (95%) using microwave assisted synthesis. The compound was characterized by FTIR, XRD, NMR, and elemental analysis. Spectrophotometric titration of 3APS was performed with Al(III), Ba(II), Cd(II), Co(II), Cu(II), Fe(III), Mn(II), Ni(II) and Zn(II). Results: 3APS showed good abilities to detect Al(III) and Fe(III) ions fluorescently and Cu(II) ion colorimetrically. The L:M stoichiometric ratio was 2:1 for Cu(II) and 1:1 for Al(III) and Fe(III). Low detection limits (μg/L) of 324, 20, and 45 were achieved for Cu(II), Al(III), and Fe(III), respectively. Detection of aluminium was also demonstrated in anti-perspirant deodorants, test strips, and applications in secret writing. Conclusions: 3APS showed high fluorescent selectivity for Al(III) and Fe(III) and colorimetric selectivity towards Cu(II) with detection limits lower than corresponding safe drinking water guidelines.

Comment #3. Drawings (Fig. 5 and 9) need to be redone.

Authors reply #3: the issues in Figures 5 and 9 are now fixed (the lines were removed).

Reviewer 2 Report

Here are my comments.

1. Overall Figure quality is really low. Fig. 5a has a weird arrow at the left and Fig. 9a has no graph line. Fig 13 has (a) and (b) half-blocked. 

2. I find no meaning in presenting chapter 2.14 synthesis of crystals. Could authors justify in the manuscript why is 3APS-Alum crystal is needed here?

3. The authors claimed that 5 min synthesis of 3APS is very promising compared to previously reported methods in terms of time. Has this method never been done by anyone but by the authors for the first time? If so, why others have not tried this method which looks relatively simple? Any critical difference between 5 min synthesis and previous methods?

4. Authors said their 5 min synthesis of 3APS was 95% yield. Any data for that?

5. Cu is detected colorimetrically. Al and Fe are detected fluorescently. Since Al and Fe share sample mechanisms for the detection, they may not be easy to be differentiated from each other. Although the authors presented L+Al+Fe fluorescent image in Fig 10b, I could not find any fluorescent photo toward Fe. Could authors also provide Fe version photos of Fig 10b for clarification?

Author Response

Replies to the last reviewer's comments are included in the attached file (last page).

Reviewer 3 Report

Dear Editor,

 I have read the manuscript entitled: “3-Aminopyridine salicylidene: a sensitive and selective 2 chemosensor for the detection of Cu(II), Al(III), and Fe(III) 3 with application to real samples ” and the following should also be checked and modified. 

Lines 94,95: The authors say “So, the method of synthesis that is developed in the current 94 work (section 3.2) is superior to the methods already reported in the literature.”  They can explain why the method in this paper is superior to methods in the literature. Lines 102,103: 3APS is repeated in the same sentence, without meaning. / In the caption of the Figure 4b the authors could also add the wavelengths corresponding to absorbances (A0 and A) used in the Benesi-Hildebrand plot. Please check the figure 5a, things seem unclear. What does nmA mean (in the figure 7), please correct? Please check figure 9a, it is not clear.  The authors say: "The concentrations of metal ion solutions added to the 3APS-Al(III) solution were 10 times higher "please check. Figure 11a (X axis): wavelength=wavelength; Figure 13 (the caption): commertial samples =commercial samples;

Minor points: line 52: prolong exposure = prolonged exposure; line 65: recognition to biologically =recognition to biologically ; line 97: synthetic route= Synthetic route; line 127: metal ions solutions = metal ion solutions; line 213: 3APS vs. Al(III) = 3APS with Al(III); line 292: the interfere was = the interference was’ ; line 313: utilizes in secret = utilized in secret

Author Response

Many thanks for the reviewers for the time and efforts they took to revise the manuscript.

Reviewer #2

I have read the manuscript entitled: “3-Aminopyridine salicylidene :a sensitive and selective 2 chemosensor for the detection of Cu(II),Al(III), and Fe(III) 3 with application to real samples ” and the following should also be checked and modified.

Comment #1. Lines 94,95: The authors say “So, the method of synthesis that is developed in the current work (section 3.2) is superior to the methods already reported in the literature.” They can explain why the method in this paper is superior to methods in the literature.

Authors reply #1: the sentence of concern has been modified to “So, the method of synthesis that is developed in the current work (section 3.2) is superior to the methods already reported in the literature due to the short synthesis time (5 minutes compared to 15-60 minutes heating or refluxing) without the need for waiting overnight for the product to appear as reported in the literature”. See the revised manuscript, lines 112-115.

Comment #2. Lines 102,103: 3APS is repeated in the same sentence, without meaning.

Authors reply #2: This comment may be related to Lines 97-98 as no 3APS mentioned in Lines 102-103. Anyhow the sentence in Lines 97-98 was modified to “To study the solvent effect on the absorbance of the sensor, solutions of 1.0 × 10 -3 M of 3APS in ethanol, acetonitrile (ACN), methanol, and water were prepared, and their absorbance spectra were measured.” See the revised manuscript, lines 117-119.

Comment #3. In the caption of the Figure 4b the authors could also add the wavelengths corresponding to absorbances (A0 and A) used in the Benesi-Hildebrand plot.

Authors reply #3: Thanks for the reviewer for noting this point. A 0 and A were taken at 385 nm. The caption of Figure 4b has been accordingly amended. Similar amendment was made for Figure 8b (405 nm).

Comment #4. Please check the figure 5a, things seem unclear.

Authors reply #4: Figure 5a was fixed.

Comment #5. What does nmA mean (in the figure 7),please correct?

Authors reply #5: Figure 7 was fixed, “A” was extra and already removed.

Comment #6. Please check figure 9a, it is not clear.

Authors reply #6: Figure 9a was fixed.

Comment #7. The authors say: "The concentrations of metal ion solutions added to the 3APS-Al(III) solution were 10 times higher "please check.

Authors reply #6: Yes, the ratio of Al(III) to each of the other metals ions was 1:10. The sentence was slightly amended to “The concentration of each of the other metal ions added to 3APS-Al(III) solution was 10-fold higher”. See the revised manuscript, lines 210-211.

Comment #8. Figure 11a (Xaxis): wavelength=wavelength

Authors reply #8: apology for this typo, it is fixed now.

Comment #9. Figure 13 (the caption): commertial samples =commercial samples;

Authors reply #9: : apology for this mistake, it is fixed now.

Comment #10. Minor points: line 52: prolong exposure = prolonged exposure; line 65: recognition to biologically =recognition to biologically ; line 97:synthetic route= Synthetic route; line 127: metal ions solutions =metal ion solutions; line 213: 3APS vs. Al(III) = 3APS with Al(III);line 292:the interfere was = the interference was’ ; line 313: utilizes in secret = utilized in secret.

Authors reply #10: thanks for the reviewer and apology for these mistakes, all have been

fixed now.

Round 2

Reviewer 2 Report

I recommend publication of the manuscript.